# CCR6 Deficiency Increases Infarct Size after Murine Acute Myocardial Infarction

**DOI:** 10.3390/biomedicines9111532

**Published:** 2021-10-25

**Authors:** David Schumacher, Elisa A. Liehn, Anjana Singh, Adelina Curaj, Erwin Wijnands, Sergio A. Lira, Frank Tacke, Joachim Jankowski, Erik A.L. Biessen, Emiel P.C. van der Vorst

**Affiliations:** 1Institute for Molecular Cardiovascular Research (IMCAR), RWTH Aachen University, 52074 Aachen, Germany; dschumacher@ukaachen.de (D.S.); acuraj@ukaachen.de (A.C.); jjankowski@ukaachen.de (J.J.); erik.biessen@mumc.nl (E.A.L.B.); 2Department of Anesthesiology, University Hospital, RWTH Aachen University, 52074 Aachen, Germany; 3Department of Intensive Care and Intermediate Care, University Hospital, RWTH Aachen University, 52074 Aachen, Germany; eliehn@ukaachen.de; 4Department of Cardiology, Angiology and Intensive Medicine, University Hospital Aachen, 52074 Aachen, Germany; 5National Institute for Pathology “Victor Babes”, 050096 Bucharest, Romania; 6Institute for Molecular Medicine, University of Southern Denmark, 5230 Odense, Denmark; 7Department of Pathology, Cardiovascular Research Institute Maastricht (CARIM), Maastricht University Medical Centre, 6229 ER Maastricht, The Netherlands; anjana2102@yahoo.co.in (A.S.); erwin.wijnands@mumc.nl (E.W.); 8Cognizant Technology Solutions, Phase II Hinjawadi, Pune 411 057, Maharashtra, India; 9Precision Immunology Institute, Icahn School of Medicine at Mount Sinai, New York, NY 10029, USA; sergio.lira@mssm.edu; 10Department of Hepatology and Gastroenterolgy, Campus Virchow-Klinikum and Campus Charité Mitte, Charité–Universitätsmedizin Berlin, 13353 Berlin, Germany; frank.tacke@charite.de; 11Interdisciplinary Center for Clinical Research (IZKF), RWTH Aachen University, 52074 Aachen, Germany; 12Institute for Cardiovascular Prevention (IPEK), Ludwig-Maximilians-University Munich, 80336 Munich, Germany; 13DZHK (German Centre for Cardiovascular Research), Partner Site Munich Heart Alliance, 80336 Munich, Germany

**Keywords:** acute myocardial infarction, ischemia-reperfusion injury, chemokine receptors, CCR6

## Abstract

Ischemia-reperfusion injury after the reopening of an occluded coronary artery is a major cause of cardiac damage and inflammation after acute myocardial infarction. The chemokine axis CCL20-CCR6 is a key player in various inflammatory processes, including atherosclerosis; however, its role in ischemia-reperfusion injury has remained elusive. Therefore, to gain more insight into the role of the CCR6 in acute myocardial infarction, we have studied cardiac injury after transient ligation of the left anterior descending coronary artery followed by reperfusion in *C**cr**6**^−/−^* mice and their respective C57Bl/6 wild-type controls. Surprisingly, *C**cr**6**^−/−^* mice demonstrated significantly reduced cardiac function and increased infarct sizes after ischemia/reperfusion. This coincided with a significant increase in cardiac inflammation, characterized by an accumulation of neutrophils and inflammatory macrophage accumulation. Chimeras with a bone marrow deficiency of CCR6 mirrored this adverse *C**cr**6**^−/−^* phenotype, while cardiac injury was unchanged in chimeras with stromal CCR6 deficiency. This study demonstrates that CCR6-dependent (bone marrow) cells exert a protective role in myocardial infarction and subsequent ischemia-reperfusion injury, supporting the notion that augmenting CCR6-dependent immune mechanisms represents an interesting therapeutic target.

## 1. Introduction

Acute myocardial infarction (AMI), caused by coronary artery occlusion resulting in dysfunctional myocardium, is one of the leading causes of death worldwide [1]. The current standard therapies are reperfusion strategies for restoring blood flow [2], although these may result in paradoxical cardiomyocyte dysfunction and worsen tissue damage in a process called ischemia-reperfusion injury [3]. This ischemic cell death eventually results in the formation of post-AMI cardiac fibrosis, leading to the formation of a fibrotic scar [4].

AMI triggers a temporally defined inflammatory response that is mediated by various leukocyte subsets [5,6]. The occurring cell death triggers the recruitment of neutrophils and monocytes. The recruitment of these inflammatory cells is controlled by chemokines that are released by the ischemic heart. Several chemokines, such as CCL2 and CXCL12, have been shown to be involved in the AMI response and the resulting cardiac necrosis and fibrosis [7,8]. Although it has been well described that chemokines and chemokines receptors are main regulators of leukocyte trafficking during immune surveillance and inflammation [9], it seems like they can have opposing effects on AMI [8]. For example, the silencing of CCR2 resulted in decreased infarct inflammation in mice subjected to AMI due to the attenuated recruitment of inflammatory Ly-6C^high^ monocytes [10], reflecting the detrimental role of CCR2 in AMI. On the other hand, the conditional deletion of endothelial ACKR3 (previously known as CXCR7) resulted in an exacerbated heart function impairment after AMI and increased infarct sizes, reflecting a protective role of ACKR3 in AMI [11]. It could also be shown that CCL5 plays a role in AMI [12,13,14], reducing cardiac injury/inflammation by reducing neutrophil and inflammatory monocyte infiltration. 

One of the chemokines that has recently been associated with ischemic heart disease, particularly AMI, is CCL20. It could be shown that in patients with an AMI, serum levels of CCL20 are significantly higher compared to those of healthy controls [15]. Furthermore, it could be demonstrated that inflamed human peripheral blood mononuclear cells have an increased expression of CCL20 [16]. Interestingly, it could also be shown that the receptor for CCL20, the chemokine receptor CCR6, plays an essential role in the recruitment of T cells to ischemic brain tissues [17]. CCR6 is highly expressed on lymphocytes, a cell type that has also been shown to contribute to ischemia-reperfusion injury after AMI [18,19]. As the role of CCR6 in AMI and cardiac injury, while plausible, is elusive, we set out to investigate this further in mice with or without CCR6 deficiency in a transient left anterior descending coronary artery ligation model and found that CCR6 deficiency increases the infract size, followed by decreased ejection fraction and stroke volume. Furthermore, CCR6 deficiency increases apoptosis and the infiltration of inflammatory cells in the injured heart. 

## 2. Materials and Methods

### 2.1. Coronary Ischemia-Reperfusion Surgery

All animal experiments were performed in accordance with European legislation and approved by local German authorities (AZ:84-02.04.2013.A185 and AZ: 8.67-50.10.35.09.088). All mice were housed under standardized conditions in the Animal Facility of the University Hospital Aachen (Germany).

Male C57Bl/6 wild-type (Janvier Labs, Le Genest-Saint-Isle, France) and *C**cr**6**^−/−^* (generously donated; described in [20]) 10–12 weeks old mice were subjected to coronary ischemia-reperfusion injury, as described previously [21,22]. For bone marrow (BM) experiments, mice underwent ischemia-reperfusion injury six weeks after BM reconstitution. Briefly, mice were anaesthetized using 100 mg/kg of ketamine and 10 mg/kg of xylazine i.p., then were intubated and positive-pressure ventilated with oxygen and 0.2% isoflurane using a mouse respirator (Harvard Apparatus, March-Hugstetten, Germany). Buprenorphine (0.1 mg/kg) was administrated 30 min prior to surgery. We performed a skin and muscle incision above the left chest and opened the chest by cutting the intercostal muscle between the second and the third rib. Then, the mice were subjected to acute myocardial ischemia-reperfusion injury by inducing left anterior descending coronary artery occlusion for 60 min using a 7/0 silk ligature. The ligature was reopened after 60 min to allow reperfusion of the myocardium. The ribs, muscle layer, and skin incision were closed, and buprenorphine (0.1 mg/kg) was administered until full recovery was achieved. 

### 2.2. Bone Marrow Transplantation

BM transplantation was performed as described previously [23]. In brief, C57Bl/6 wild-type or *C**cr**6**^−/−^* eight-week-old male mice were irradiated and reconstituted with either C57Bl/6 wild-type or *C**cr**6**^−/−^* BM. Femurs and tibias were aseptically removed from donor C57Bl/6 wild-type or *C**cr**6**^−/−^* mice and BM cavities were flushed for BM isolation. Single-cell suspensions were prepared and donor cells (2 × 10^6^) were administered to the recipient mice by tail-vein injection 24 h after an ablative dose of whole-body irradiation (2 × 6 Gray). The BM chimeras were split into four different groups, wild-type mice reconstituted with wild-type BM, wild-type mice reconstituted with *C**cr**6**^−/−^* BM, *C**cr**6**^−/−^* mice reconstituted with wild-type BM and *C**cr**6**^−/−^* mice reconstituted with *C**cr**6**^−/−^* BM. Mice were left to recover for 2 weeks, after which they were subjected to transient coronary artery ligation to be sacrificed 3 weeks after AMI.

### 2.3. Echocardiography

The left ventricular heart function was determined by echocardiography performed on a small-animal ultrasound imager (Vevo 770, FUJIFILM Visualsonics, Toronto, ON, Canada) before and three weeks after coronary ischemia-reperfusion injury. Measurements of short and long cardiac axis were taken in B-Mode (2D-realtime) and M-Mode using a 40 MHz transducer. During the procedure, mice were anesthetized with 1–2% isoflurane. The ejection fraction (EF %), stroke volume (µL) and left ventricular diameters (mm) were recorded and analyzed using the VevoLab Software (FUJIFILM Visualsonics, Toronto, ON, Canada).

### 2.4. Histology and Immunohistochemistry

Infarct size was evaluated 3 weeks after coronary ischemia-reperfusion injury. Mice were euthanized by isoflurane overdose and their hearts were excised, fixed in formalin, and embedded in paraffin. Serial sections (10–12 sections per mouse, 400 µm apart, up to the mitral valve) were stained with Gomori’s 1-step trichrome stain. The infarcted area was determined for all sections using Diskus software (Hilgers, Königswinter, Germany) and expressed as a percentage of total left ventricular volume, as described before [24]. Serial sections (3 sections per mouse, 400 µm apart) 2 days after coronary ischemia-reperfusion were stained to analyze the infarcted area for proliferation (KI67, ThermoFisher Scientific, Schwerte, Germany), apoptosis (TUNEL stain, In Situ Cell Death Detection Kit, Sigma-Aldrich, Darmstadt, Germany), reparatory macrophages (Mac3, BD Pharmingen, Heidelberg, Germany), inflammatory macrophages (Mac3 and MPO double positive; Mac3, BD Biosciences, Heidelberg, Germany; MPO, Neomarkers, ThermoFisher Scientific, Schwerte, Germany), neutrophils (MPO, Neomarkers, ThermoFisher Scientific, Schwerte, Germany), cardiomyocytes (Troponin, ThermoFischer Scientififc, Schwerte, Germany), and B-cells (CD45R/B220, BD Biosciences, Heidelberg, Germany). Serial sections from 21 days after coronary ischemia-reperfusion injury were stained for myofibroblasts (smooth muscle actin, DAKO, Jena, Germany) and neoangiogenesis (CD31, Santa Cruz Biotechnology, Dallas, Texas, USA). Positive-stained cells or vessels were counted in six different fields per section and expressed as number of cells per mm^2^.

### 2.5. Statistical Analysis

Data represent mean ± SD. Statistical analysis was performed with the Prism7 software (GraphPad, San Diego, CA, USA). The means of two groups were compared with unpaired Student’s t test, using Welch’s correction by significant variance. More than two groups were analyzed using 1-way ANOVA followed by Tukey’s multiple comparison test. *p*-values of < 0.05 were considered significant.

## 3. Results

### 3.1. CCR6 Deficiency Worsens Cardiac Function and Increases Infarction Size after Ischemia-Reperfusion Injury

To investigate the role of CCR6 in AMI and cardiac injury, we performed coronary ischemia-reperfusion surgery in C57Bl/6 wild-type and *C**cr**6**^−/−^* mice. Heart function was assessed three weeks after injury to measure early lesion formation and the recovery of cardiac performance (Figure 1A). The mortality rate of *C**cr**6**^−/−^* mice was strongly increased from week 2 onwards, in comparison to wild-type controls (87.5% vs. 50% for wild-type controls, respectively) (Figure 1B). The ejection fraction and stroke volume were significantly decreased in *C**cr**6**^−/−^* mice (20% and 30%, respectively, compared to wild-type controls), whereas the left ventricular end-systolic volume was strongly, though not significantly, increased (Figure 1C−E). Furthermore, in *C**cr**6**^−/−^* mice the infarction size was increased by 70% compared to wild-type mice (Figure 1F,G). Together, these data clearly indicate that deficiency of CCR6 worsens cardiac function after ischemia-reperfusion injury.

### 3.2. Bone Marrow Derived Immune Cells Are Responsible for the Effects of Ccr6^−/−^ after Coronary Ischemia-Reperfusion Injury

To further assess whether resident cells or bone marrow cells are responsible for the increased infarction size and worsened heart function in *C**cr**6**^−/−^* mice, we performed bone marrow transplantation experiments (Figure 2A). Mice receiving *C**cr**6**^−/−^* BM showed a clear trend (15%) towards a reduced ejection fraction compared to mice receiving BM from wild-type mice (Figure 2B). The genotype of the recipient mice (*C**cr**6**^−/−^* vs. wild type) had no effect on the ejection fraction. In line with this, mice receiving *C**cr**6**^−/−^* BM had a significantly increased infarction size compared to controls receiving BM from wild-type mice (3.76% ± 0.44 vs. 2.49% ± 0.33 for *Ccr6**^−/−^* recipients and 3.60% ± 0.95 vs. 2.36% ± 0.57 for wild-type recipients; Figure 2C,D). Conversely, the genotype of the recipient mice (*C**cr**6**^−/−^* vs. wild-type) again had no significant effect (3.76% ± 0.44 vs. 3.60% ± 0.95 for *C**cr**6**^−/−^* BM and 2.49% ± 0.33 vs. 2.36% ± 0.57 for WT BM; Figure 2C,D). The non-significant though clear effect on ejection fraction can thereby be explained by the relative small infarct size. Combined, these results clearly demonstrate that bone marrow CCR6, rather than stromal CCR6, is responsible for the observed profound effects on infarct size. 

### 3.3. Increased Cardiac Infiltration of Inflammatory Immune Cells in CCR6^−/−^ Mice

Decreased cardiac function following coronary ischemia-reperfusion injury can be caused either by increased inflammation and cell death, the exacerbation of the adverse remodeling by myofibroblasts or decreased vessel formation. To clarify the exact reason for the increased infarction size in *C**cr**6**^−/−^* mice, we first assessed the myofibroblast content and neovascularization in the infarct tissue, 21 days after ischemia-reperfusion, which both were not affected by CCR6 deficiency (Figure 3A,B). To elucidate the effects of CCR6 deficiency on inflammation, 2 days after coronary ischemia-reperfusion injury the number of B-cells, neutrophils, and inflammatory macrophages was analyzed, demonstrating that *C**cr**6**^−/−^* mice demonstrate more inflammatory cell infiltration into the tissue compared to their wild-type littermates (Figure 3C−G), suggesting that increased inflammation is a major underlying cause for the observed increased infarction size in these mice. In line with this notion, the number of reparative macrophages in the tissue was significantly decreased in *C**cr**6**^−/−^* mice compared to wild-type controls (Figure 3E), suggesting insufficient reparatory processes after AMI.

Furthermore, we determined the degree of cellular proliferation and apoptosis in the cardiac tissue 2 days after coronary ischemia-reperfusion injury. TUNEL staining showed that CCR6 deficiency led to an almost 5-fold, albeit not significant increase in apoptosis (Figure 4A−C), which could mainly be observed in cardiomyocytes (Figure 4B), while Ki−67 staining, especially present in myofibroblasts, demonstrated that proliferation was unaffected (Figure 4D−F). 

## 4. Discussion

In this study, we aimed to elucidate the role of CCR6 in AMI and subsequent cardiac injury. We were able to show that CCR6 deficiency, more specifically in bone marrow cells, has detrimental effects on cardiac function and infarct size after ischemia-reperfusion injury. This coincided with a strong increase in cardiac tissue inflammation, characterized by the presence of inflammatory cells such as neutrophils and inflammatory macrophages, and impaired reparatory processes due to reduced reparatory macrophages. 

The role of CCL20-CCR6 axis in AMI has scarcely been studied, but can be inferred from the contribution of CCR6 expressing subsets to infarct responses in the heart and brain [17,18,19,25,26]. B cells play a key contributory role in ischemia-reperfusion injury and cardiac inflammation [27]. The genetic and antibody-mediated depletion of mature B cells resulted in reduced myocardial injury and monocyte infiltration, thereby improving heart function [27]. In that study, the increased monocyte mobilization and recruitment to the heart was linked to the increased secretion of CCL7 by B cells. Indeed, we did observe B-cell accumulation in the cardiac tissue of *Ccr6**^−/−^* mice, albeit it that this effect did not reach significance, probably due to the small number of animals analysed.

Further, there are data supporting the role of CCR6 in B cell development. Intriguingly, it could be observed that the lack of CCR6 led to the increased presence of the appearance of germinal centres in the spleen and increases the maturation of B cells, although the produced antibodies in *C**cr**6**^−/−^* B cells were of lower affinity [28,29]. In addition, CCR6 uniquely marks memory B cells (and precursors) in both mouse and human germinal centres, is essential for the appropriate anatomical positioning of memory B cells, and coordinates antigen recall by these cells [30,31]. Strikingly, B cells were also seen to migrate to ischemic areas in the brain in response to injury [25], which corresponds to the increased B-cell presence in ischemic heart tissue seen in our study.

CCR6 is also known to play an important role in T cell trafficking. Both Th17 and regulatory (Treg) cells express CCR6, which has been linked to the susceptibility to autoimmune diseases [26]. Interestingly, CCR6 expressed by tissue resident γδ T cells has also been shown to play an important role in stroke and subsequent brain ischemia [17]. A lack of CCR6 in Th17 cells has also been shown to inhibit their recruitment [26]. It seems that Th17 cells also interact with neutrophils during inflammatory processes stimulating the production of IL-23/Th17-associated cytokines, which leads to severe tissue damage and impaired repair in psoriasis [32], acute kidney injury [33], infective endocarditis [34], or multiple sclerosis [35]. Furthermore, the role of the CCR6/CCL20 axis has also already been elaborately investigated in other autoimmune diseases such as inflammatory bowel disease and rheumatoid arthritis, which also highlights its importance in Th17 cells, as recently reviewed in [36]. Strikingly similar to our AMI model, CCR6-deficient mice show aggravated tissue injury, macrophage/neutrophil inflammation, and fibrosis in models of chronic liver injury. In the liver, CCR6 is necessary for the accumulation of the subset of interleukin (IL)-17- and IL-22-expressing γδ T cells that restricts hepatic inflammation and fibrosis [37]. In the heart, Toll-like receptor signaling and IL-1β, present during neutrophil infiltration [38,39], together with IL23, expand IL-17A production [40]. IL-17A seems to act in the late phase of remodeling by increasing cardiomyocyte injury, promoting the sustained infiltration of proinflammatory cells, and enhancing fibroblast proliferation [40]. 

A lack of CCR6 in Th17 cells also limits the recruitment of Treg’s into inflammatory tissues in a Th17-dependent manner. Similarly, Treg expressed CCR6 also plays a crucial role in their migratory response to sites of inflammation [26]. In the healthy myocardium, only few resident Treg cells are present, however they rapidly infiltrate the tissue in the context of AMI [41,42]. Several studies have reported a beneficial role of Tregs in AMI [19]. For example, Treg expansion improves both survival and myocardial wound healing by shifting macrophages toward a more reparatory and fibrotic phenotype [43]. 

Supportive of a role in AMI, a recent study demonstrated that AMI patients have significantly more circulating CCR6^+^ lymphocytes [44] and its ligand CCL20 [15] compared to controls, rendering the CCR6/CCL20 axis a potential biomarker of AMI. However, both studies involved rather small cohorts, warranting the further confirmation of these findings in a larger cohort. Furthermore, the former study was designed as a cross-sectional study and was therefore unable to demonstrate any causality between CCR6 and AMI. By inference, it is clear that CCR6 deficiency could impact on AMI response, possibly by modulating B cell/monocyte interaction or Treg mobilisation, although the exact underlying mechanisms remain to be elucidated in future studies.

## 5. Conclusions

In this study, we demonstrate that a deficiency in CCR6 and, more specifically, a deficiency in CCR6 expressed in bone marrow cells aggravates cardiac function loss in a model of cardiac ischemia-reperfusion injury, which is accompanied by the sustained accumulation of B-cells, neutrophils, and inflammatory macrophages as well as a reduction in the resident macrophage pool. Though it is tempting to speculate that these effects are caused by the reduced accumulation of Treg or reparative B cell subsets, future studies are needed to fully elucidate the exact underlying mechanisms and the therapeutic potential of CCR6/CCL20 targeting in AMI. Furthermore, the investigation of the importance of the CCL20/CCR6 axis in AMI should be performed in larger human cohorts for validation purposes.

## Figures and Tables

**Figure 1 biomedicines-09-01532-f001:**
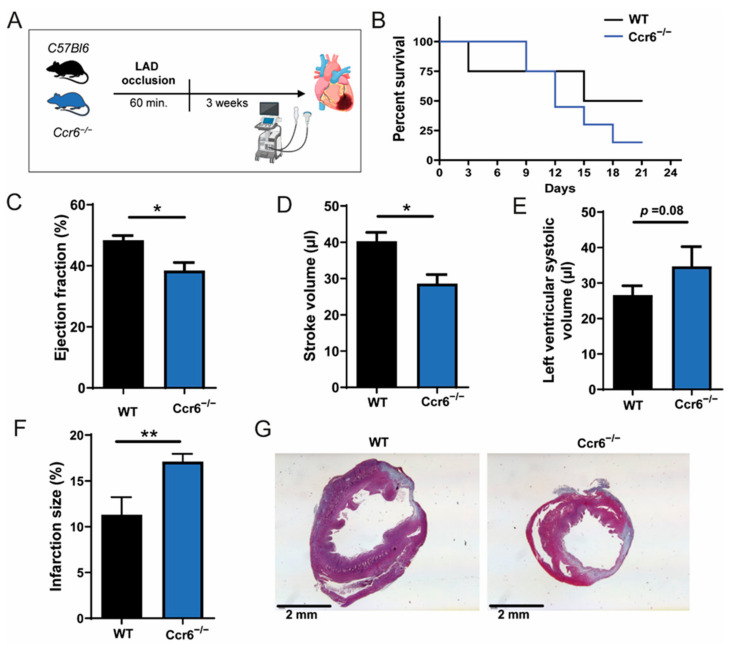
CCR6 deficiency results in decreased cardiac function and increased infarction size after ischemia-reperfusion injury. (**A**) Study design: wild-type and *Ccr6**^−/−^* mice were subjected to ischemia-reperfusion injury, echocardiography was performed 3 weeks after injury (*n* = 3). Image was created using Biorender.com (Accessed on 2 September 2021). (**B**) Survival curve after ischemia-reperfusion injury. (**C**) Ejection fraction in *Ccr6**^−/−^* mice compared to wild-type controls. (**D**) Stroke volume in *Ccr6**^−/−^* mice compared to wild-type controls. (**E**) End-systolic volume in *Ccr6**^−/−^* mice compared to wild-type controls. (**F**) Infarction size in *Ccr6**^−/−^* mice compared to wild-type controls. (**G**) Representative images of Gomori’s 1-step trichrome stain. * *p* < 0.05, ** *p* < 0.01.

**Figure 2 biomedicines-09-01532-f002:**
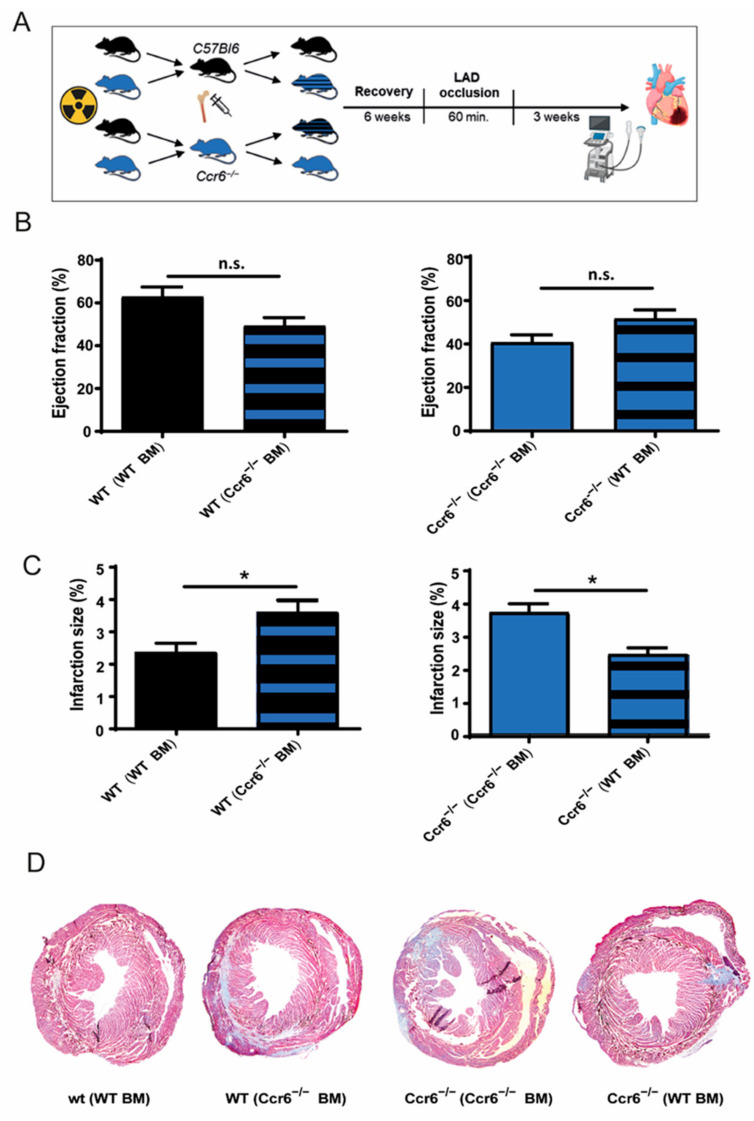
CCR6 deficiency in the bone marrow results in increased infarction size after ischemia-reperfusion injury. (**A**) Study design of bone marrow transplantation experiments, where the mice were subjected to ischemia-reperfusion injury. Echocardiography was performed 3 weeks after injury. Image was created using Biorender.com (Accessed on 2 September 2021). (**B**) Ejection fraction in wild-type recipient mice receiving either wild-type or *C**cr**6**^−/−^* BM (*n* = 4–6) and *C**cr**6**^−/−^* recipient mice receiving either wild-type or *C**cr6^−/−^* BM (*n* = 3). (**C**) Infarction size in wild-type recipient mice receiving either wild-type or *C**cr6^−/−^* BM (*n* = 4–6) and *C**cr**6**^−/−^* recipient mice receiving either wild-type or *C**cr**6**^−/−^* BM (*n* = 3). (**D**) Representative images of heart sections stained with Gomori’1 step trichrome staining of wild-type recipient mice receiving either wild-type or *C**cr**6**^−/−^* BM and *C**cr**6**^−/−^* recipient mice receiving either wild-type or *C**cr**6**^−/−^* BM. * *p* < 0.05.

**Figure 3 biomedicines-09-01532-f003:**
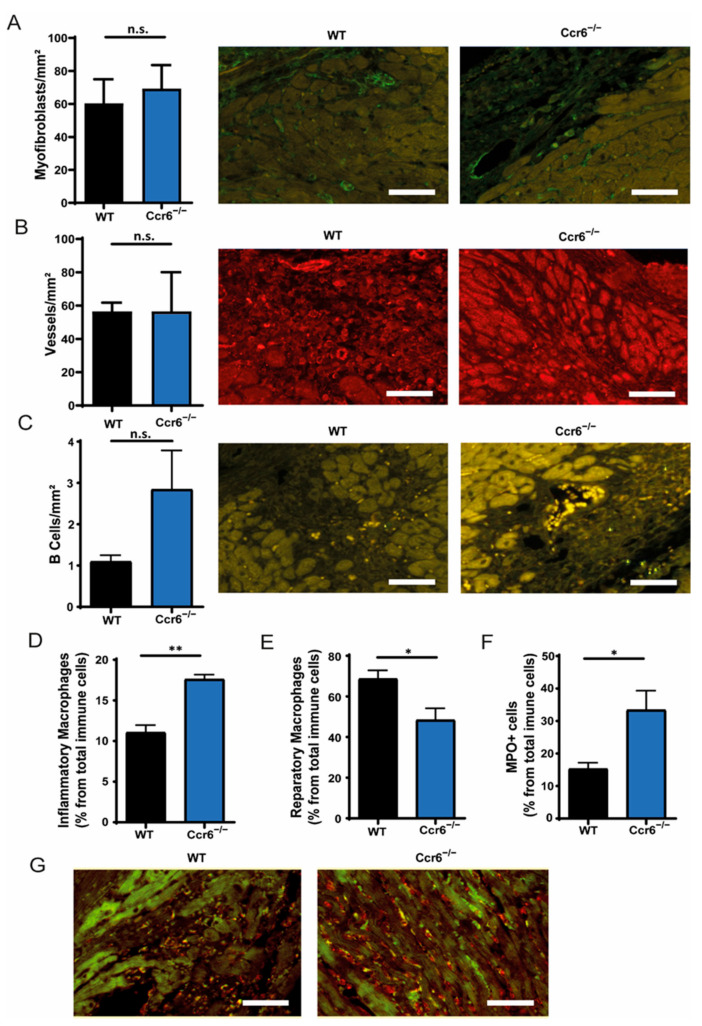
Deficiency of CCR6 results in the increased infiltration of inflammatory immune cells in the cardiac tissue after ischemia-reperfusion injury. (**A**) Quantification of the number of myofibroblast in the scar from *C**cr**6**^−/−^* mice compared to wild-type mice 21 days after ischemia-reperfusion, including representative images of smooth-muscle actin (Green) staining. Scale bar = 50 µm. (**B**) Quantification of the number of vessels in the scar from *C**cr**6**^−/−^* mice compared to wild-type mice 21 days after ischemia-reperfusion, including representative images of CD31 (Red) staining. Scale bar = 50 µm. (**C**) Quantification of infiltrating B cells, 2 days after ischemia-reperfusion in *C**cr**6**^−/−^* mice compared to wild-type mice, including representative images of B220 (Yellow) staining. Scale bar = 50 µm. (**D**–**F**) Quantification of infiltrating inflammatory MPO+ immune cells (**D**), inflammatory macrophages (**E**) and reparatory macrophages (**F**), 2 days after ischemia-reperfusion in *C**cr**6**^−/−^* mice compared to wild-type mice. (**G**) Representative images of MPO (Red) and Mac3 (Green) staining. All *n* = 3. * *p* < 0.05, ** *p* < 0.01.

**Figure 4 biomedicines-09-01532-f004:**
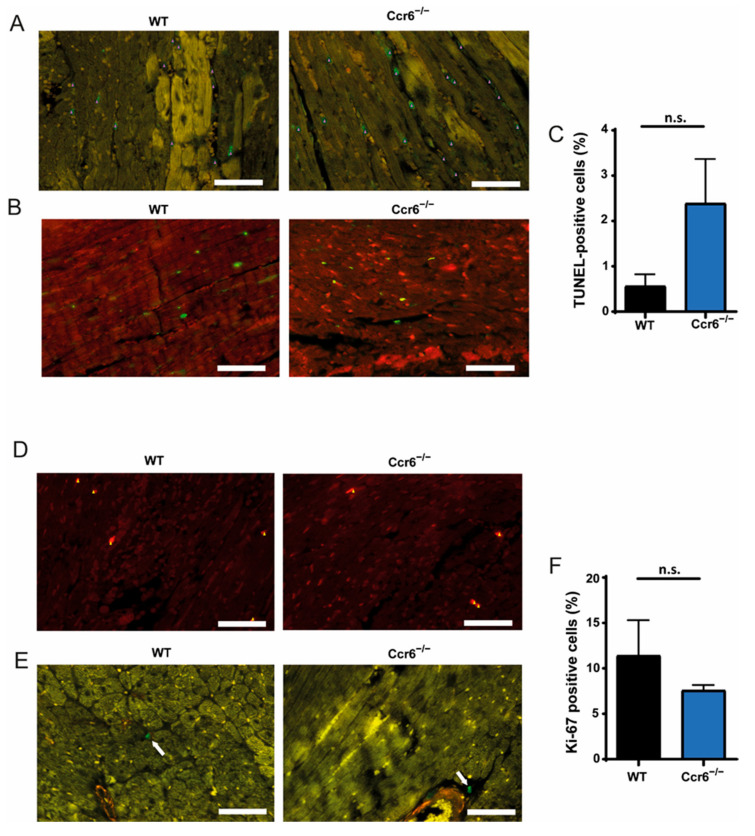
Deficiency of CCR6 results in increased apoptosis, especially in cardiomyocytes, after ischemia-reperfusion injury. (**A**–**C**) Representative images of TUNEL (Green) staining (**A**) and TUNEL (Green)/Troponin (Red) double staining (**B**), including the quantification of total TUNEL+ cells (**C**), 2 days after ischemia-reperfusion in *C**cr**6**^−/−^* mice compared to wild-type mice. Scale bar = 50 µm. (**D**–**F**) Representative images of Ki67 (Green) staining (**D**) and Ki67 (Green)/smooth-muscle actin (Red) double staining (**E**), including the quantification of total Ki67+ cells (**F**), 2 days after ischemia-reperfusion in *C**cr**6**^−/−^* mice compared to wild-type mice. Scale bar = 50 µm. Arrows point at positively stained cells. All *n* = 3.

## Data Availability

All data presented in this study are available upon reasonable request from the corresponding author.

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
