# Peer review of "CCR6 Deficiency Increases Infarct Size after Murine Acute Myocardial Infarction"

_biomedicines, 2021, doi:10.3390/biomedicines9111532_

Round 1
Reviewer 1 Report
- Please indicate where were the mice purchased from, especially Ccr6 -/- mice.
- Fig 1. I would suggest the authors separate Fig 1 into 2 figures, in respect of results 3.1 and 3.2.
- For result 3.2, could the authors show the data of echo, survival rate, and representative figures of Masson's staining.
- For fig 2, could the authors show representative figures of all the panels.
- I noticed the authors showed no statistical significance in TUNEL and Ki67, albite there were a obvious tendency of difference. I would suggest the authors to do TUNEL and ki67 staining together with EC marker (e.g. CD31) and CM marker (e.g. cTnT), respectively, to show the apoptosis and proliferation in each cell type. Also, PH3 could be also used for proliferation detection.
- Although CCR6 plays an important role in T cells and is is highly expressed on lymphocytes, I don't think the authors can draw a conclusion that haematopoietic CCR6 deficiency is the reason for cardiac function lost. The authors used a globally Ccr6 KO mouse instead of hematopoietic conditional KO, in spite of using the BM transplantation model, the data still could not rule out the role of BM stromal stem cells play in ischemic heart repair. In my opinion, it would be more appropriate to use the term "Ccr6 deficient bone marrow cells"
- The title, quote "CCR6+ haematopoetic cells restrict infarct size". However, the manuscript only proved CCR6 -/- increased infarct size, not the other way around. We don't know if by transplanting CCR6+ haematopoetic cells in the MI heart could preserve the heart function or not. Please reconsider this title.
- If the authors really want to focus on the protective role of CCR6+ haematopoetic cells, I would suggest the authors isolate haematopoetic cells from WT mice, the inject the cells into CCR6 -/- mice, to rescue the worsened heart function.
Reviewer 2 Report
This study demonstrated in mice model that CCR6-dependent hematopoietic (immune) cells exert a protective role in myocardial infarction and subsequent ischemia-reperfusion injury, supporting that augmenting CCR6-dependent immune mechanisms represents an interesting therapeutic target. Study is interesting and innovative, as there is little evidence in other publications on the presented subject.
However, as Authors searched for novel links between myocardial ischemia and inflammation and extent of post-myocardial injury, Authors should provide more rationale for choosing this chemokine receptor. Whether, serum levels of CCL20, higher in AMI, link to the chemokine receptor CCR6 in the heart or serum? Is it a true link or coincidence? Further studies with higher number of samles should be performed. For that reason Authors should give their conclusions with caution.
CCR6+ was studied in the context of autoimmune disease, e.g. colitis, SLE, rheumatitis, where Th helpers and IL-17 levels has gained wide interest as treatment target.
Authors should also provide an information on methods of injury-reperfusion, even if it was already described in former publications.
Round 2
Reviewer 1 Report
The authors have addressed all my previous concerns. I do not have more questions.
Reviewer 2 Report
Authors answered all comments. Please finish sentence in discussion , as it ends with in ... [36].